# Uncovering the Worldwide Diversity and Evolution of the Virome of the Mosquitoes *Aedes aegypti* and *Aedes albopictus*

**DOI:** 10.3390/microorganisms9081653

**Published:** 2021-08-03

**Authors:** Rhys Parry, Maddie E James, Sassan Asgari

**Affiliations:** 1School of Chemistry and Molecular Biosciences, The University of Queensland, Brisbane, QLD 4072, Australia; 2School of Biological Sciences, The University of Queensland, Brisbane, QLD 4072, Australia; m.james4@uq.edu.au (M.E.J.); s.asgari@uq.edu.au (S.A.)

**Keywords:** insect viruses, *Aedes aegypti*, *Aedes albopictus*, ISV, virome, Jingmen tick virus, *Wolbachia*

## Abstract

*Aedes aegypti*, the yellow fever mosquito, and *Aedes albopictus*, the Asian tiger mosquito, are the most significant vectors of dengue, Zika, and Chikungunya viruses globally. Studies examining host factors that control arbovirus transmission demonstrate that insect-specific viruses (ISVs) can modulate mosquitoes’ susceptibility to arbovirus infection in both in vivo and in vitro co-infection models. While research is ongoing to implicate individual ISVs as proviral or antiviral factors, we have a limited understanding of the composition and diversity of the *Aedes* virome. To address this gap, we used a meta-analysis approach to uncover virome diversity by analysing ~3000 available RNA sequencing libraries representing a worldwide geographic range for both mosquitoes. We identified ten novel viruses and previously characterised viruses, including mononegaviruses, orthomyxoviruses, negeviruses, and a novel bi-segmented negev-like group. Phylogenetic analysis suggests close relatedness to mosquito viruses implying likely insect host range except for one arbovirus, the multi-segmented Jingmen tick virus (*Flaviviridae*) in an Italian colony of *Ae. albopictus*. Individual mosquito transcriptomes revealed remarkable inter-host variation of ISVs within individuals from the same colony and heterogeneity between different laboratory strains. Additionally, we identified striking virus diversity in *Wolbachia* infected *Aedes* cell lines. This study expands our understanding of the virome of these important vectors. It provides a resource for further assessing the ecology, evolution, and interaction of ISVs with their mosquito hosts and the arboviruses they transmit.

## 1. Introduction

The yellow fever mosquito *Aedes aegypti*, and the Asian tiger mosquito *Aedes albopictus*, are invasive hematophagous insects distributed worldwide within the tropical and subtropical zones [1]. Both species have expanded from ancestral forest niches into peri-domestic settings facilitated by artificial containers and global shipping routes [2,3]. 

Laboratory studies of *Ae. aegypti* and *Ae. albopictus* have shown that both are competent vectors of numerous arthropod-borne viruses (arboviruses) such as yellow fever virus (YFV), Zika virus (ZIKV), and dengue virus (DENV) (reviewed in [4,5]). Chikungunya virus (CHIKV) (*Togaviridae*) is also competently vectored by both mosquitoes [6]. DENV is the most widespread arbovirus worldwide, ubiquitous in the tropics, and recently introduced into Europe [7]. Estimates of DENV infection suggest up to 390 million annual cases, of which ~100 million manifests into various disease severity levels [8]. While both *Ae. aegypti* and *Ae. albopictus* can vector DENV, reviews of vector competence between the two species indicated that DENV dissemination rates are lower in *Ae. albopictus* [9], which is considered a secondary or maintenance vector relative to *Ae. aegypti* [5]. While ZIKV and YFV are estimated to cause fewer annual cases than DENV, both viruses have been responsible for devastating outbreaks in endemic regions, placing billions of people at risk [10,11]. During the 2015–2016 Central and South American ZIKV outbreak, there were an estimated 400,000 cases of ZIKV [12]. Experimental determination of vector competence for both these species for ZIKV suggests that *Ae. aegypti* is a superior vector to *Ae. albopictus* [13]. YFV is endemic in regions of up to 47 countries in Africa and Central and South America, with approximately 200,000 annual infections resulting in at least 30,000 fatalities [14]. 

Vector competence encompasses all the intrinsic and extrinsic host factors and the interplay between these factors (reviewed in [15]). Identifying host factors that modulate vector competence to arboviruses in mosquitoes is an attractive area of research. Exploiting these mechanisms offers the potential to curb the burden of arbovirus disease [16]. Extrinsic or environmental factors such as host-associated microbiota can profoundly impact the vector competence of mosquitoes (reviewed in [17]). Due to arbovirus surveillance programs, several viruses isolated from *Ae. aegypti* and *Ae. albopictus* have been discovered, which do not productively infect vertebrate cells and only replicate in mosquito or insect cells. These insect-specific viruses (ISVs) are unlikely to exist in a vector-borne transmission cycle and are maintained through horizontal or vertical transmission. The presence of ISVs can suppress or enhance the subsequent replication of arboviruses. Experimentally ISV-infected *Ae. aegypti* cells modulate the ability of cells to produce DENV-2 [18,19]. Many in vivo and in vitro studies in *Ae. aegypti* and *Ae. albopictus* have shown that ISV-arbovirus interference can be from the same virus family as the arbovirus [20,21,22,23] or completely different virus groups [24,25,26]. The mechanism(s) of this interference phenomenon remains the subject of ongoing research. 

Virus discovery generally takes either a culture-dependent or culture-independent approach. In culture-independent methods, samples are subjected to high-throughput total RNA sequencing (RNA-Seq), allowing extensive resolution of the ecology and incidence of viruses. These techniques have been used extensively across eukaryotic systems [27,28,29,30]. Library preparation for metatranscriptomics studies typically involves the clearance or depletion of ribosomal RNA and is not enriched for poly-A mRNA transcripts, a library preparation step typically used for differential gene expression analysis. While the poly-A enrichment step does bias the population of viruses identified, with sufficient depth, many non-poly-A enriched viruses can also be assembled and identified from these data.

In addition to total-RNA sequencing, small RNA (16–32 nt) sequencing is also especially useful for virus discovery; as during virus infection and replication, the host endoribonuclease III enzyme Dicer-2 processes dsRNA produced by viruses during replication into short interfering RNAs (siRNAs) between 20–22 nt length with a strong bias for 21 nt [31,32,33]. Collectively, this pathway is referred to as the RNA interference (RNAi) response and can promote tolerance to virus infection. Furthermore, active RNAi response targeting both strands of a virus genome suggests strong evidence for ongoing replication of viruses, and *de novo* assembly using these sRNAs has allowed for the recovery of virus contigs [34,35].

As we slowly appreciate the role the virome can play in vector competence, we sought to uncover the incidence, diversity, and evolution of these viruses in *Ae. aegypti* and *Ae. albopictus* from previously published RNA-Seq data. We included cell lines, laboratory strains, and wild-caught mosquito pools in our analyses. The outcomes provide a comprehensive list of ISVs present in these mosquitoes and their diversity between and within mosquito populations. 

## 2. Materials and Methods

### 2.1. Collation of Metadata from Ae. aegypti and Ae. albopictus Virus Publications

To identify publications utilising high-throughput RNA-Seq of *Ae. aegypti* and *Ae. albopictus*, we searched Web of Science (http://www.webofknowledge.com; accessed on 20 April 2020) and PubMed (https://www.ncbi.nlm.nih.gov/pubmed; accessed on 20 April 2020) using the following search terms: (“*Aedes*”[Title] OR “*Aedes*”[All Fields]) AND (RNA-Seq[Title] OR transcriptomic[Title] OR transcriptome[Title] OR sequencing[Title]). We cross-referenced these publications with raw data deposited in Sequence Read Archive (SRA) hosted by NCBI. We excluded libraries and publications where metadata was not usable, conflicted with methods described within the publication or if uploaders of data were uncontactable. Original catch locations of colonies latitude and longitude positions were approximated from metadata and plotted using ggplot2 (H. Wickham, 2016) and the package “maps” (v 3.3.0, https://CRAN.R-project.org/package=maps; accessed on 20 April 2020) in the RStudio environment (Version 0.99.491). Fastq files from associated SRA accession numbers were downloaded using a Perl script written by Michael Gerth (Oxford Brookes University) and available from GitHub (https://github.com/gerthmicha/perlscripts/blob/master/sra_download.pl; accessed on 20 April 2020). 

### 2.2. Virus Discovery Pipeline 

For virus discovery, we split computational tasks evenly across the Galaxy workbench (version 18.09) and CLC Genomics workbench (12.0.3). RNA-Seq fastq files were downloaded, and FastQC (Version 0.11.8, https://www.bioinformatics.babraham.ac.uk/projects/fastqc/; accessed on 20 April 2020) was used to identify adapters. Adapters and low-quality regions of sequencing were trimmed using the CLC in-house trimming tool (quality = 0.05, ambiguous nucleotides = 2) or Trimmomatic (galaxy version: 0.36.4) [36] under the following conditions (Sliding window = 4, average quality = 20). *De novo* assembly was then undertaken using CLC Genomics Workbench under default conditions or using the Trinity Galaxy wrapper (version 2.4.0.2) [37]. To identify novel viruses, we created a local BLAST database from virus protein sequences deposited in the non-redundant NCBI database (NCBI:txid10239). We excluded several large dsDNA virus families to reduce computational requirements, owing to a high proportion of false-positive BLASTx hits with host and phage proteins. The resulting file (n = 581,356) was collapsed for similarity using the CD-hit tool (Version 4.7) [38] under default conditions (n = 62,840). Assembled contigs were then queried against the representative virus protein database using BLASTx (Expect: 10, Word size: 3, Matrix BLOSUM45, Gap cost Existence 15, Extension 2) [39]. To incriminate viruses as bona fide, we considered several criteria: completeness of coding regions, evidence of sub-genomic transcription, and conserved virus protein domains. For multi-segmented viruses, all segments had to co-appear in multiple libraries or within sequencing libraries from individual mosquitoes; a reciprocal BLASTn analysis of independently assembled virus strains from different libraries allowed to identify chimeric virus assemblies. Putative virus strains were re-mapped to RNA-Seq data to inspect for sufficient coverage and possible misassembly.

As endogenous viral elements (EVEs) are abundant in mosquito nuclear genomes and transcriptionally active [40,41], we excluded the possibility that virus contigs are derived from these EVEs by identifying retrotransposable elements or also partial fragments of totally unrelated viruses, a feature of viral EVE clusters [40]. As PIWI-RNAs (piRNAs) against a single orientation of a viral genome with no virus-derived siRNA (vsiRNA) response in mosquitoes is a classic demonstration of endogenisation of a virus, this was also used to exclude “viruses” that are potentially endogenised. Finally, we screened all the resultant viruses with the recent assemblies of the *Ae. aegypti* (AaegL5) and *Ae. albopictus* Foshan (AaloF1) genomes through BLASTn analysis.

### 2.3. Virus Genome Annotation

Assembled contigs were subjected to BLASTx analysis using our curated database to identify contigs bearing high identity to previously described viruses. Hits bearing an expected value (*E-*value) < 10^−5^ were manually inspected for chimeric contigs and then annotated. We predicted the open reading frames of these viruses using the NCBI Open Reading Frame Finder (https://www.ncbi.nlm.nih.gov/orffinder/; accessed on 20 April 2020) with a minimum ORF length of 150 aa and using standard genetic code. Translated ORFs were then analyzed for putative domains with the NCBI Conserved Domain Search (https://www.ncbi.nlm.nih.gov/Structure/cdd/wrpsb.cgi; accessed on 20 April 2020) and cross-referenced with the Pfam protein database (v 33.1) search (https://pfam.xfam.org; accessed on 20 April 2020). Predicted ORFs with no known similarity to any proteins based on BLASTp analysis were also analyzed using PSI-BLAST to identify highly divergent homologs [42]. Predicted transmembrane domain helices and signal peptides were identified using the TMHMM Server v. 2.0 (http://www.cbs.dtu.dk/services/TMHMM/; accessed on 20 April 2020) and using the SignalP 3.0 Server (http://www.cbs.dtu.dk/services/SignalP-3.0/; accessed on 20 April 2020), respectively. Nuclear Localisation Signals were predicted using the cNLS Mapper under default conditions [43].

### 2.4. Virus Coverage Statistics and Virus Incidence Heat Maps

To obtain coverage statistics and to validate potential misassemblies, clean fastq reads were mapped using BWA-MEM [44] and visually inspected using the Integrative Genomics Viewer (IGV v2.6.2) [45]. Coverage was calculated from the resultant Binary Alignment (BAM) files using bedtools [46] based on 5′ positions and graphed using GraphPad Prism (v9.0.0). For the representation of the incidence of viral RNA composition between RNA-Seq libraries, adapter, and quality trimmed fastq libraries were mapped against a representative virus database using the CLC Genomics Workbench mapping program (coverage = 0.9, identity = 0.9). Total mapped reads are presented as heat maps using the pheatmap package (v 1.0.12) (https://cran.r-project.org/web/packages/pheatmap/; accessed on 20 April 2020) in the RStudio environment (Version 0.99.491). 

### 2.5. Analysis of Virus-Derived Small RNAs in Aedes

For examination of the viral RNAi response, we used a previously described pipeline [47]. Briefly, adapter and quality trimmed fastq files (>18 nt) from small RNA datasets were mapped to virus genomes using Bowtie2 (Galaxy Version 2.3.4.3) [48] and coverage calculated as above. Extracted sequence signatures of 27 nt virus-derived piRNAs (vpiRNAs) were generated using Weblogo3 from mapped sense and reverse-complemented antisense reads. Sequence overlaps of vpiRNAs were determined using the small RNA signatures tool version 3.1.0 hosted on the Mississippi Galaxy server (https://mississippi.snv.jussieu.fr; accessed on 20 April 2020) with 24–29 nt viral mapped reads as input.

### 2.6. Phylogenetic Analysis of Aedes Viruses

For the phylogenetic placement of novel viruses within respective families, we used predicted RNA-dependent RNA polymerase (RdRp) proteins. Closely related viruses from BLASTp analysis of the NCBI non-redundant protein database were aligned using MUSCLE [49]. Ambiguously aligned regions were removed from the alignment using TrimAl (v. 1.3) under the automated 1 method [50]. For phylogenetic inferences of viruses, maximum likelihood trees were produced using IQ-TREE (v1.6.10) [51] using the Le-Gascuel (LG) model [52] with discrete gamma model with 4 rate categories (+G4) and 50,000 Ultrafast bootstraps [53]. FigTree version 1.4 (A. Rambaut; https://github.com/rambaut/figtree/releases; accessed on 20 April 2020) was used to visualise consensus trees.

## 3. Results and Discussion

### 3.1. Many Novel Viruses Are Associated with Aedes aegypti and Aedes albopictus Colonies and Cell Lines

To discover and profile viruses infecting *Ae. aegypti* and *Ae. albopictus*, we downloaded high-throughput RNA-Seq libraries deposited in the Sequence Read Archive (SRA) database (Figure 1A). These samples represent tissues from various laboratory colonies lab-adapted from various geographic locations or wild-caught outright (Figure 1B, Appendix A) and cell line RNA-Seq data produced from both species. In total, we identified up to 10 novel viruses (Table 1). BLASTp analysis of predicted novel virus sequences based on the RdRp genes suggests viruses presented here are closely related to those known to infect mosquitoes or other insect species (Table 1). 

### 3.2. Novel Negative-Sense RNA Viruses Infecting Ae. aegypti and Ae. albopictus

We identified three novel negative-strand mononegaviruses (Appendix A for details). The first, Formosus virus, a *Rhabdovirus* from a lab colony of *Ae. aegypti formosus was* established from wild-caught samples from Bundibugyo, Uganda [54]. Formosus virus shared close pairwise amino acid similarity to the Culex rhabdo-like virus assembled from a metagenomic analysis from *Culex* mosquitoes in California, USA [55] (Figure 2B). Phylogenetic placement of Formosus virus from alignments of the L protein suggests Formosus virus groups within *Rhabdoviridae*. 

All members of *Mononegavirales* have a negative-stranded RNA genome encapsidated within the capsid and the RNA polymerase complex [56]. The RNA polymerase complex sequentially transcribes discrete mRNAs from the genome with mRNA from each gene, which are then capped and polyadenylated. Importantly, all three viruses had sub-genomic mapping profiles typical of mononegaviruses from poly-A enriched libraries (Appendix A)

We identified a multi-segmented negative-sense RNA virus from the *Orthomyxoviridae* family (Figure 3) infecting *Ae. albopictus* samples, which we name Aedes orthomyxo-like virus 2 (AOMV-2). This virus is very lowly abundant in Foshan *Ae. albopictus* RNA-Seq data [69,70] but abundant in RNA-Seq data from a lab colony established from the Torres Strait in Australia [72,123]. Phylogenetic placement of this virus based on alignments of the RdRp polymerase basic 1 subunit (PB1) places AOMV-2 within a well-supported mosquito clade of unassigned orthomyxoviruses (Figure 3B). Orthomyxoviruses have a nuclear replication strategy, and we predicted several monopartite and bipartite nuclear localisation signals on multiple proteins from AOMV-2, showing a close association with the host nucleus (Appendix A). 

### 3.3. Novel Positive-Sense RNA Negev-like Viruses Infecting Ae. aegypti and Ae. albopictus

Negeviruses are positive-sense RNA viruses with genome sizes between 9–10 kb encoding three ORFs that infect multiple mosquito species and sandflies. All negeviruses currently reported have been unable to replicate in vertebrate cell cultures [124]. We identified one novel coding complete negev-like virus (Figure 4A; details in Appendix A) in *Ae. aegypti* originating from two colonies collected in Rabai, Kenya (K2, K4), and therefore, we named the virus Rabai virus [54]. Phylogenetic analyses of the Rabai virus based on the highly conserved VMet, HEL, and RdRp domains suggests a close relatedness to other negev-like viruses assembled from metagenomic studies of mosquitoes (Figure 4B) but does not belong to the monophyletic *Nelorpivirus* and *Sandewavirus* taxons as previously reported [125].

### 3.4. Binegeviruses: A Novel Negev-Related Taxon with Bi-Segmented Genomes in Aedes Mosquitoes

In addition to the monopartite negev-virus genomes, we discovered two novel bi-segmented viruses distantly related to the negev-like viruses in both *Ae. aegypti* and *Ae. albopictus* (Figure 4A). The larger segment of these “binegeviruses” is ~7600 nt, encodes for two ORFs with the prototypical negevirus viral methyltransferase (VMet) and helicase (HEL) domains (Appendix A). The second smaller segment is ~4600 nt and contains two ORFs with an RdRp. TBLASTn analysis of the predicted proteins of both segments suggests that these viruses are most closely related to the 7413 nt ssRNA virus-like sequence 6 (Genbank ID: KX148585) and the 4642 nt ssRNA virus-like sequence 5 (Genbank ID: KX148584) identified from metagenomics analysis of *Anopheles* mosquitoes [126]. Phylogenetic inferences based on alignments of the VMet, HEL, and RdRp domains from both viruses, tentatively named Aedes binegev-like virus 1 (AeBNV-1) and Aedes binegev-like virus 2 (AeBNV-2), suggest the formation of a phylogenetically divergent and well-supported grouping distinct from other negev-viruses (Figure 4B) and grouping with a partial RdRp from Blackford virus identified from *Drosophila tristis* pools [127]. Both segments of AeBNV-1 and AeBNV-2 co-appear in all mosquito pools positive for these viruses in numerous American *Ae. aegypti* datasets and individual *Ae. albopictus* mosquitoes from Longgang District, Shenzhen, China [68], supporting the likelihood of these viruses belonging to the same group. For this reason, we believe it is likely that the ssRNA virus-like sequence 5/6 previously reported belongs to the same virus as both co-appeared in pools of mosquitoes from Senegal and Liberia [126]. Bi-segmentation and tri-segmentation of viruses from the *Virgaviridae* family are well established [128]. The RdRp_2 domain on a smaller segment is a classic feature of the plant infecting *Hordeivirus* genus with the barley stripe virus as the best-known member [128].

### 3.5. Tombus-Noda Viruses in Aedes

Previously, fragments of a potential “Mosquito nodavirus” were assembled from small RNA-Seq data from one Liverpool colony of *Ae. aegypti* [34,35]. We were able to complete the genome of this virus by assembling multiple RNA-Seq libraries from numerous Liverpool *Ae. aegypti* datasets and find that this “Mosquito nodavirus” is not segmented like classical nodaviruses but instead has a classical monopartite 4 kb genome (Figure 5A). The virus is more closely related to the single genome Tombus-Noda arthropod lineage (Figure 5B) [73]. For this reason, we chose to re-name this virus Liverpool tombus-like virus, as it appears to infect Liverpool colonies of *Ae. aegypti* exclusively, with one exception being one *Ae. albopictus* colony co-housed with the *Ae. aegypti* Liverpool strain [129]. Additionally, we identified a divergent strain of a similar tombus-noda virus from the Foshan *Ae. albopictus* colony, which we named Foshan tombus-like virus, with similar genome features and phylogenetic position (Figure 5B). We identified a common novel multi-segmented tombus-like virus, which is a ubiquitous infectious agent in *Ae. albopictus* mosquitoes, termed tiger mosquito bi-segmented tombus-like virus (TMTLV). The genome orientation and structure of TMTLV is very similar to Diaphorina citri-associated C virus (DcACV) from the Asian citrus psyllid (*Diaphorina citri*) to which TMTLV is phylogenetically related (Figure 5B) [130], with segment one on TMTLV and DcACV encoding the RdRp_3 domain. The second segment encodes a putative DiSB-ORF2_chro domain, a putative virion glycoprotein of insect viruses (Appendix A) [131]. This virus is ubiquitous in American, Asian, and European *Ae. albopictus* mosquitoes (Table 1). TMTLV is abundantly targeted by the RNAi response in the Foshan colonies and *Ae. albopictus* strain at the University of Pavia [93,94], and recently lab-adapted strains from Vietnam and Japan [59,86].

### 3.6. Double-Stranded RNA Viruses Infecting Ae. aegypti and Ae. albopictus

Partitivirus-like sequences belonging to a putative partitivirus named Aedes partiti-like virus 1 (APLV-1) were distributed worldwide in *Ae. aegypti* colonies (Table 1, Figure 6A). Phylogenetic placement of the RdRp segment of APLV-1 indicated a close relationship of APLV-1 and mosquito and dipteran partitiviruses (Figure 6B). Members of the *Partitiviridae* family possess two genome segments, dsRNA1 (RdRp) and dsRNA2 (Virus Coat), each containing one large ORF between 1.4–2.4 kb. Each genome segment is separately encapsidated. Some partitiviruses have additional (satellite or defective) dsRNA elements. We identified two segments of a putative partiti-like virus in three Rabai, Kenya (K2, K4, K14) colonies [54]. Another colony from this study, the K27 colony, did not have any reads mapping to APLV-1 but was crossed with the APLV-1 positive colony K14. We identified both segments in all the libraries in each of the four progeny libraries (GP1, GP2, HP1, HP2). This suggests that APLV-1, like the recently identified partitivirus Verdadero virus in *Ae. aegypti* mosquitoes may exhibit efficient vertical transmission. Using the massively high-throughput nature of our metanalysis, we could incriminate the Chaq-like virus (Genbank ID: MT742176.1), previously identified in Verdadero virus-infected *Ae. aegypti* samples, as a likely satellite or additional segment of Verdadero virus. Chaq-like virus (Genbank ID: MT742176.1) sequences co-appeared in every library along with the two segments of Verdadero virus from multiple independent sequencing efforts (11/11 BioProject accessions) and importantly co-appear in up to eight individual mosquitoes from individual midgut libraries produced by Raquin et al., 2017 [119], although in low abundance. We identified a virus belonging to the *Reoviridae* family that is phylogenetically related to the *Orbivirus* genus (Figure 6C,D), which we have tentatively named Aedes orbi-like virus. Phylogenetic analysis of the VP1 segment of the virus suggests a close relationship to orbiviruses assembled from a pool of *Ochlerotatus* mosquitoes (Figure 6D) [132]. In total, we were able to identify seven segments designated VP1-7 as per orbivirus convention (Figure 6C). 

While four of the AOLV segments had detectable homologs from other insect reo-like viruses, three out of the seven incriminated segments of AOLV had no detectable homology with any known virus from this group or RNA viruses (Appendix A), suggesting a highly divergent genome within this virus taxon. Identification of these additional segments of AOLV was aided by co-appearance in numerous individual transcriptomes from a field adapted colony from Cairns, Australia [108]. In addition to co-appearance in individual mosquito transcriptomes, all seven segments are targeted by the RNAi response in lab colonies from Far North QLD, Australia [105].

### 3.7. DNA Viruses of Aedes Mosquitoes

Mosquito densoviruses (MDVs) are single-stranded DNA viruses of the *Parvoviridae* family known to infect both *Ae. aegypti* and *Ae. albopictus* (reviewed in [133]). MDVs in *Ae. aegypti* and *Ae. albopictus* belong to the type species *Dipteran brevidensovirus* 1 and 2 of the monosense *Brevidensovirus* genus [134]. We identified several datasets of *Ae. albopictus* colonies and cell lines infected with MDVs and were able to recover the complete coding genome (NS1/2 and VP genes) of these putative MDV strains. Phylogenetic placement of the NS1/2 and VP genes of these contigs (Figure 7) suggested these MDVs are most closely related to MDVs previously identified from other mosquitoes. Based on species demarcation criteria of the *Parvoviridae* (<80% amino acid identity from the replicase protein), they are not sufficiently divergent to be separate species of MDV.

After the initial assembly of a putative MDV from the *Ae. albopictus* U4.4 cells sequenced as part of the Arthropod Cell Line RNA-Seq initiative, Broad Institute, we examined the incidence of this MDV sequence in U4.4 cells and could not identify reads mapping to this MDV from any other U4.4 datasets (Appendix A). Given this densovirus is most closely related to Anopheles gambiae densovirus (AgDV), which has been previously isolated from the *Anopheles gambiae* cell line Sua5B [135], we presume that it is likely sequencing contamination from Sua5B cells. Further examination of all *Aedes* cells from this initiative demonstrates that the virus variant is abundant in all *Aedes* cell lines but absent in other U4.4 datasets (Appendix A). Additionally, we identified one *Ae. albopictus* C6/36 sRNA dataset that produces abundant vpiRNAs against another more distantly related MDV, Culex pipiens densovirus (CppDNV). We were even able to partially assemble the virus genome from this data [136]. Examination of the incidence of CppDNV in other C6/36 cells suggested that CppDNV was absent in every C6/36 RNA-Seq dataset examined (Appendix A). This indicates that either CppDNV is so lowly abundant that it is beyond detection, potentially integrated into the genome, or only exists in a handful of laboratory C6/36 isolates. In addition to potential infection of *Aedes* cells, we identified MDV infections in *Ae. albopictus* RNA-Seq data from Manassas, USA [90] and the Foshan, China colony (62, 63). The coding-complete genome of MDV from the *Ae. albopictus* Foshan colony (BioProject: PRJNA275727) is most closely related to Aedes albopictus densovirus 7 isolate GZ07 (Figure 7) [137].

### 3.8. Evidence of the Jingmen Tick Virus (Flaviviridae) in an Ae. albopictus Mosquito Colony

In 2014, Jingmen tick virus (JMTV), a segmented positive-sense RNA virus, was reported following isolation from *Rhipicephalus microplus* tick and mosquito pools collected in the Jingmen region of Hubei Province, China [138]. The genome structure of JMTV comprises four polyadenylated segments, two of which share close homology to non-structural (NS) proteins of classical flaviviruses, with segment one encoding an NS5-like RdRp and methyltransferase, and segment three bearing close similarity to the protease (NS3), with the other two segments originating from a yet unidentified ancestor (Figure 8A). JMTV replicates in *Ae. albopictus* C6/36 and several mammalian cell lines [138,139]. Since the first description of JMTV, there has been evidence of human infections from both Kosovo [140] and China [139]. We assembled an almost complete JMTV strain from the Rimini, Italy colony of *Ae. albopictus* except for a 27 nt gap in segment 1. Subsequent re-mapping of all libraries from this study indicated that only three out of the 24 libraries had detectable JMTV RNA (BioProject PRJNA493544: SRA Accessions: SRR7907917, SRR7907927, SRR7907936) [141]. Phylogenetic analysis of the conserved Segment 1 and Segment 3 of the Rimini JMTV strain suggests that all segments are most closely related to JMTV strains from ticks sampled from the French Antilles [142] and also Trinidad and Tobago [143] (Figure 8B,C), sharing approximately 92–95% nucleotide identity over the coding genome.

These RNA-Seq data samples are replicate pools of 10–24 fat bodies and heads from 5-day-old females fed on 20% sucrose and water. Given the tissues sampled and that JMTV reads were lowly abundant in these libraries (0.02% of all reads), it appears unlikely that the introduction of JMTV into these samples are due to recent blood-feeding for maintenance. However, given that the Rimini colony was established in 2004 and sampled in November 2017, we cannot rule out the possibility that JMTV may have been introduced from the previous blood-feeding to maintain the colony. Further examination of the incidence and persistence of JMTV in *Aedes* mosquitoes is required.

### 3.9. Diversity of the RNAi Response against ISVs of Ae. aegypti and Ae. albopictus

To examine the RNAi response to *Aedes* ISVs, we mapped reads from small RNA sequencing libraries against our virus database, examining the size distribution of the virus fragments originating from these genomes and the mapping profiles of *Aedes* viruses (Extended Figures in Appendix A). We demonstrate that ten of these viruses presented with prototypical virus-derived short interfering RNAs (vsiRNAs) as part of the RNAi response with a 21-nucleotide peak targeting both sense and antisense genomes. A summary of the size distribution profile of mapped reads to virus genomes is presented in Figure 9. 

The virus-derived small RNA profiles are consistent with published virus RNAi profiles from closely related viruses; for example, San Gabriel mononegavirus has abundant virus-derived P-element Induced Wimpy testis RNAs (vpiRNA) length small RNAs dominant in the population of small RNAs consistent with other studies examining the RNAi response of mononegaviruses [18,144]. 

We could identify the vsiRNA profile for both segments of the Verdadero virus, a partiti-like virus previously identified in *Ae. aegypti* colony metatranscriptomics [145]. We also show that the positive-sense RNA virus Wenzhou sobemo-like virus 4 produces an RNAi response in vivo in *Ae. albopictus* mosquitoes from several libraries [86,87]. We could also identify the RNAi signature from both Renna virus segments [29] (Figure 9). 

In addition to virus RNA being targeted and degraded by the RNAi pathway, we also observed that many of these viruses abundantly produce virus-derived RNAs of size 24–30 nt corresponding to vpiRNAs. vpiRNAs can be determined by a ping-pong signature (U_1_-A_10_) and 10 nt complementary overlapping reads. For the San Gabriel virus and all segments of Aedes phasma virus [58,146], as well as individual segments of TMTLV and WSBLV-4, we observed a ping-pong signature (U_1_-A_10_) and over-represented complementary 10 nt overlapping pairs of sRNA from reads of 24–29 nt from these viruses (Appendix A). The absence of a predominant 21 nt species of vsiRNA targeting CppDNV and WSBLV-4 in *Ae. albopictus* C6/36 and C7-10, respectively, are consistent with these cell lines RNAi deficient nature. Virus infections within these cell lines do not produce a typical 21 vsiRNA peak. Instead, dsRNA is targeted and processed in C6/36 and C7-10 cells by the piRNA pathway [129,147]. 

### 3.10. Differentially Abundant ISVs in Aedes Laboratory Colonies 

To understand the composition and incidence of ISVs between different laboratory colonies, we created a database of representative genomes of novel viruses identified here and previously identified viruses known to infect these mosquitoes (n = 94). Libraries originating from one biological sample or group were trimmed, pooled, and mapped to these genomes. Previous studies have used an arbitrary cut-off of 100 mapped reads per million using a total library sub-sample. Some of the viruses presented here either do not have poly-A tails or are unlikely to have poly-A tails. As most of the data examined were produced from transcriptome studies using a poly-A enrichment step, we felt this cut-off might be too stringent. Instead, we used the complete libraries and visually inspected mapping coverage to exclude false-positive mapping. To additionally implicate viruses as being present, we queried our *de novo* assemblies through BLASTn analysis. We manually assessed hits (*E-*value < 10^−5^) as bearing >85% nucleotide identity over a minimum of 500 bp for each virus. It is essential to appreciate that the RNA samples processing, sequencing library preparation, instruments used, and read length of all data used between studies are vastly different. As such, it is challenging to normalise and draw quantitative comparisons between studies. We believe, however, that the analysis presented here reasonably incriminates samples as positively infected. However, we concede that given the limitations of the library preparations, viruses not containing poly-A tails or in huge abundance may be incorrectly characterised as “negative”. Given this limitation, we recommend screening mosquito samples using RT-PCR or RT-qPCR methods. A summary of common ISV infections and their associated BioProject is available in Appendix A.

Examining the general trends between the ISV composition of *Ae. aegypti* and *Ae. albopictus* suggests that ISV compositions are entirely separate between mosquitoes, with very few examples of potential contamination between laboratory colonies (Appendix A). Singleton virus infections in colonies are rare, and in most colonies, there exists a common suite of viruses. While the “core virome” describes common virus populations between single species in metatranscriptomics studies [121,146], multiple factors beyond host species identity influence the ecology and diversity of the metazoan virome [148]. 

In laboratory strains of *Ae aegypti* that had recently been field-adapted (within 10 generations), almost all BioProjects (57/60) had sufficient ISV RNA identifiable in these libraries, with some harbouring up to 11 individual ISVs. While recently adapted colonies of mosquitoes had the most diverse viromes, there were some *Ae. aegypti* colonies that we could not detect any virus RNA in their corresponding data and, based on our analysis, are potentially “virus sterile”. The ROCK or Rockefeller strain, which is of Caribbean origin established in 1930 [149], as well as the Rexville-D strain, established from an isofemale line collected in Rexville, Puerto Rico in the early 1990s [150], both appear to have very low, or no mapped reads originating from RNA viruses. Given that the Rexville-D strain was established from a single isofemale, it seems likely that this reduced the virome diversity.

By comparison, the Liverpool reference stain, used to generate the *Ae. aegypti* genome and maintained at the Liverpool School of Tropical Medicine since 1936 (reviewed in [149]), was identified to have almost three-quarters of BioProjects infected with Aedes aegypti toti-like virus (ATLV-1) (18/24 BioPojects) and up to a third infected with Liverpool tombus-like virus (LTLV; 7/24 BioProjects). While generally Liverpool and Rexville-D RNA-Seq libraries did not appear to be consistently infected with any other viruses, we identified a handful of examples in which specific libraries of these strains were infected with more than ATLV-1 or Liverpool virus (Appendix A). For instance, one study examining the tRNA fragments of various colonies, including the Rexville, Liverpool, and Trinidad colony, observed a convincing RNAi response to HTV and Renna virus in the Liverpool colony [120]. This suggests active infection in these samples. While lane contamination during sequencing may be responsible for this observation, it is also possible that co-housing different strains of mosquitoes in a facility might cause contamination. 

The reduction in virome diversity was similarly observed in the Foshan or Guangdong strain of *Ae. albopictus,* which has been in culture since 1981 and used to assemble the nuclear genome [69]. While Foshan colonies had a less diverse virome than recently lab-adapted strains, there was evidence of potential MDV infection in 4/8 BioProjects, and a low abundance of AOMV-2 (2/8 BioProject accessions). By comparison, laboratory colonies recently lab-adapted from wild-caught samples had the most diverse viromes [59,86].

### 3.11. Individual Ae. aegypti and Ae. albopictus Mosquitoes from the Same Colony Harbour Heterogeneous Virus Populations

Transcriptomic studies are increasingly using libraries prepared from single individuals rather than pooled mosquitoes as biological replicates. We set out to analyze individual datasets to explore the heterogeneity of viruses within individual mosquitoes. For this, we used two studies using individual *Ae. aegypti* samples recently adapted to the laboratory (within ten generations). The first study sequenced the midguts from individual mosquitoes from Thep Na Korn, Thailand midguts [119]. The second study examined 42 whole adult samples from a colony established from wild-caught mosquitoes in Cairns, Australia and sequenced after three generations [108].

Both publications experimentally infected *Ae. aegypti* with clinical isolates of dengue virus (DENV-1) [119], and (DENV-3) [108]. As the genome of DENV does not have a poly-A tail, we felt that DENV read numbers may be a reasonable detection limit for virus populations. The DENV-3 read numbers in *Ae. aegypti* Cairns libraries [108] were low but detectable in the DENV positive samples with 6–335 mapped reads ($\bar{x}$ = 90.27, n = 18). However, in the midgut libraries produced by Raquin et al. 2017 [119], at 24hpi in the individual midgut samples, DENV-1 reads were between 0–82 mapped reads ($\bar{x}$ = 18.35 n = 17), and at 96hpi there were 298–105442 reads mapped to the DENV-1 genome ($\bar{x}$ = 25,328.58, n = 17).

Within the recently colonised Thep Na Korn, Thailand strain [119], we identified reads of up to eight viruses within the midgut libraries produced for the study. Only two midgut libraries did not have any detectable reads of any virus and 15/47 midgut libraries had reads mapping to at least one virus (Figure 10A). Most individual libraries harboured multiple virus populations, with 17/47 carrying two, 12/47 carrying three viruses, and one infected with four viruses. The most common virus present in this colony was the Phasi Charoen-like virus (PCLV) with 41/47, followed by Humaita-Tubiacanga virus (HTV) (14/47) samples infected, and both Verdadero virus and Aedes anphevirus (AeAV) in 6/47 midgut samples. 

Virus heterogeneity was similar in whole mosquitoes originating from Cairns, Australia. However, the most abundant identified virus was Aedes aegypti toti-like virus 1 (ATLV-1), which was almost fixed in the mosquitoes with ~90% of individuals infected with ATLV-1 (38/42 samples) (Figure 10B). The second most abundant was PCLV (26/42), followed by HTV in 19/42 libraries and APLV-1. In this dataset, every single mosquito was infected with at least one virus with five samples with one virus, 5/42 samples with two ISVs, 13/42 samples with three ISVs, 15/42 samples with four ISVs, and 4/42 samples infected with five ISVs. These data indicate that at least in recent colonies of *Aedes* mosquitoes, most individual mosquitoes are infected on average with two to three ISVs and can be super-infected with up to five viruses in individual mosquitoes.

### 3.12. Composition of Viruses in Commonly Used Aedes Cell Lines Reveals Super-Infection of ISVs in Wolbachia Transfected Cell Lines

Cell lines are invaluable tools in arbovirus research, and numerous reports have demonstrated that most harbour persistent infections of ISVs [151]. Early records of the *Ae. aegypti* Aag2 cell line indicated a persistent infection with an insect-specific flavivirus CFAV [152,153,154]. Aag2 also persistently harbours the negative-sense PCLV (*Phenuiviridae*) [35,155], and one report of this cell line also suggests infection with AeAV [156]. PCLV is also known to infect the *Ae. albopictus* RML-12 cell line [157]. It is also well-established that the Culex Y virus (*Birnaviridae*) is a common laboratory contaminant of *Ae. albopictus* U4.4 cell line [129,151] and occasionally cross-contaminating *Ae. albopictus* C7-10 cells and *Ae. aegypti* Aag2 cells [158]. 

In addition to RNA virus infection, mosquito cell lines are known to be persistently infected with densoviruses [159]. Aedes albopictus densovirus (AalDNV-2) was first completely sequenced and characterised [160,161] from a persistently infected C6/36 cell line [162]. Previously, we have demonstrated that most published Aag2 cells appear to have a defective Aedes albopictus densovirus (AalDNV-2) genome and a truncated VP gene [163]. Defective AalDNV-2 genomes are exclusively targeted by the vsiRNA response, whereas complete AalDNV-2 infection in Aag2 cells produces both vsiRNA and vpiRNA species [163]. 

Previously reported ISVs were identified in this study (Appendix A). However, additional infections of *Aedes* cell lines were uncovered. For example, we identified Wenzhou Sobemo-like virus 4 (WSBLV-4) in *Ae. albopictus* C7-10 cells from total-RNA sequencing obtained as part of the Arthropod Cell Line RNA-Seq initiative, Broad Institute (broadinstitute.org), and three independent small RNA sequencing datasets [129,164]. It appears that not all C7-10 cells were infected with WSBLV-4; however, [58] suggesting heterogeneity between sources of C7-10 cell lines. Additionally, we identified a convincing RNAi response (mostly piRNAs) against WSBLV-4 from C6/36 cells (SRA: SRR11252294) from this dataset [58]. The RNAi profile of WSBLV-4 infection in C7-10 and C6/36 cells is in line with the Dicer-2 deficient nature of the cell line (Figure 9).

There was evidence that cell lines transinfected with the endosymbiotic bacterium *Wolbachia* are super-infected with a range of viruses, including the positive-sense RNA virus WSBLV-4 and three negative-sense RNA viruses San Gabriel virus, PCLV, and AeAV (Figure 11). 

The super-infection of WSBLV-4, San Gabriel virus, PCLV, and AeAV was observed in two independent *Ae. albopictus* C6/36 *w*MelPop-CLA infected cells [64,65] and *Ae. albopictus* RML-12 *w*MelPop-CLA infected cells [62]. San Gabriel virus infecting *Wolbachia* cell line derivatives (RML-12.*w*MelPop, Aag2.*w*MelPop, C6/36.*w*MelPop) were more closely related to the cell line-identified San Gabriel virus strains (98–99% over 99% of the genome) than to the wild-derived San Gabriel virus from 95.86% over 99% of the genome of the *Ae. albopictus* colony from the USA. This observation indicates that these infections are likely to have originated from the parental RML-12 cell lines for reasons previously discussed [18].

We have previously demonstrated that another mononegavirus, AeAV, persistently infects the RML-12 and *Ae. aegypti* Aag2.*w*MelPop-CLA cell lines with up to 1.2 million out of ~22 million small RNA reads (~5.45%) mapping to AeAV [18]. Reanalysis of these cell line data indicated a smaller fraction of vsiRNA reads from the library ~0.54% map to the San Gabriel virus genome in Aag2.*w*MelPop-CLA cells (119,424/22,078,545 reads). The RNAi profile indicates both the genome and replicative intermediate are targeted by the RNAi response, indicating active infection in these cells. By comparison, we could map no reads from the paired Aag2 cell line. We identified both WSBLV-4 and Saint Gabriel virus in the *Ae. albopictus* Aa23 cell line and a second *Ae. aegypti* cell line transcriptome that was stably transinfected with the *Wolbachia w*AlbB strain from Aa23 cells (Aag2.*w*AlbB), suggesting contamination of San Gabriel virus from the Aa23 cells during the production of the Aag2.*w*AlbB cell line [63]. The presence of three negative-sense RNA viruses is consistent with reports that *Wolbachia* does not confer a refractory virus phenotype to negative-sense RNA viruses [18,165,166]. However, examinations between *Wolbachia* and the positive sense WSBLV-4 deserve further experimentation.

## 4. Conclusions

We have identified up to ten novel viruses associated with *Ae. aegypti* and *Ae. albopictus* colonies that infect colonies, wild-caught populations, and cell lines worldwide. Phylogenetic analysis of the viruses reported here suggests all these viruses are closely related to other virus families that have recently been described to infect arthropods except for one *Ae. albopictus* colony potentially infected with the vertebrate infecting Jingmen tick virus. While most of these viruses are likely to have an insect-restricted host range, experimental validation should be undertaken due to the extensive distribution of some of these viruses in *Aedes* populations. We show that recently established lab colonies harbour up to five viruses in individual mosquitoes and very few colonies of *Aedes* mosquitoes are “sterile” using our analysis. In addition to identifying novel viruses, we have explored the geographical distribution of previously known ISVs of *Ae. aegypti* and *Ae. albopictus*. As we establish the contribution of viruses associated with mosquitoes to the vector competence of arboviruses, understanding the prevalence and distribution of these viruses provides a resource for further assessment of the ecology, evolution, and interaction of ISVs with their mosquito hosts and arboviruses they transmit.

## Figures and Tables

**Figure 1 microorganisms-09-01653-f001:**
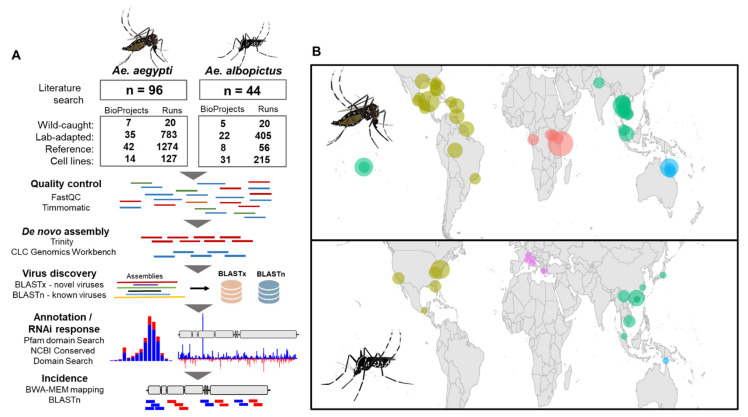
Virus discovery and geographic sample incidence in *Aedes aegypti* and *Aedes albopictus*. (**A**) The bioinformatics pipeline for virus discovery and virus incidence of *Ae. aegypti* and *Ae. albopictus* viruses. (**B**) Sample locations of *Ae. aegypti* (top) and *Ae. albopictus* (bottom). Samples are coloured based on the geographic region of the original catch location. The size of the circle indicates the number of independent catch locations and are coloured as follows: African samples (red), Australian samples (blue), Asian samples (green), American samples (gold), and European samples (pink).

**Figure 2 microorganisms-09-01653-f002:**
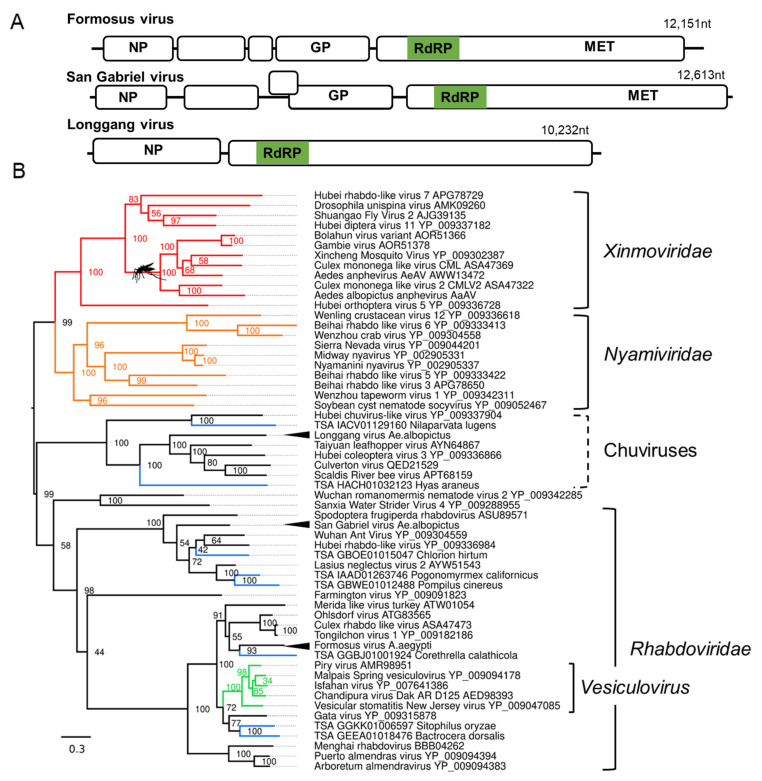
Novel negative-sense RNA viruses of *Ae. aegypti* and *Ae. albopictus* mosquitoes. (**A**) Genome structure and organisation of novel mononegaviruses. NP, nucleocapsid; GP, Glycoprotein; MET, methyltransferase. (**B**) Phylogenetic placement of novel mononegaviruses constructed based on an alignment of the L protein sequence. Branches have been coloured as per ICTV’s *Mononegavirales* taxonomy [122] with the family *Nyamiviridae* (orange), *Xinmoviridae* (red) and *Rhabdoviridae*, genus *Vesiculovirus* (green). Consensus maximum-likelihood trees were generated using IQTree with the LG + G4 amino acid substitution model with 50,000 Ultrafast bootstraps and bootstrap support indicated on the node. Each TSA accession is shown in blue. For clarity, the tree is midpoint rooted. Branch length indicates the number of amino acid substitutions per site. Novel viruses are marked with arrowheads. Nodes with a mosquito indicate clades containing viruses identified from mosquito species.

**Figure 3 microorganisms-09-01653-f003:**
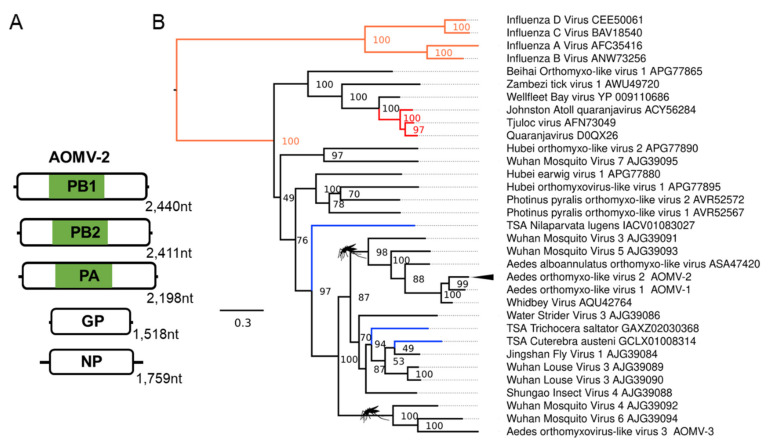
Novel negative-sense RNA viruses of *Ae. aegypti* and *Ae. albopictus* mosquitoes. (**A**) Genome structure of the Aedes orthomyxo-like virus 2 (AOMLV-2), PB1/PB2, Polymerase basic 1/2 protein; PA, polymerase acidic protein; GP, putative glycoprotein; NP, nucleoprotein. (**B**) Phylogenetic placement within *Orthomyxoviridae* constructed using aligned PB1 protein of orthomyxoviruses. Clades are coloured as per the current *Orthomyxoviridae* taxonomy (ICTV 2018b Release) with influenza viruses (orange) and genus *Quaranjavirus* (red). Consensus maximum-likelihood trees were made using IQTree with the LG + G4 amino acid substitution model with 50,000 Ultrafast bootstraps and bootstrap support indicated on the node. Each TSA accession is shown in blue. For clarity, the tree is midpoint rooted. Branch length indicates the number of amino acid substitutions per site. Novel viruses are marked with arrowheads. Nodes with a mosquito indicate clades containing viruses identified from mosquito species.

**Figure 4 microorganisms-09-01653-f004:**
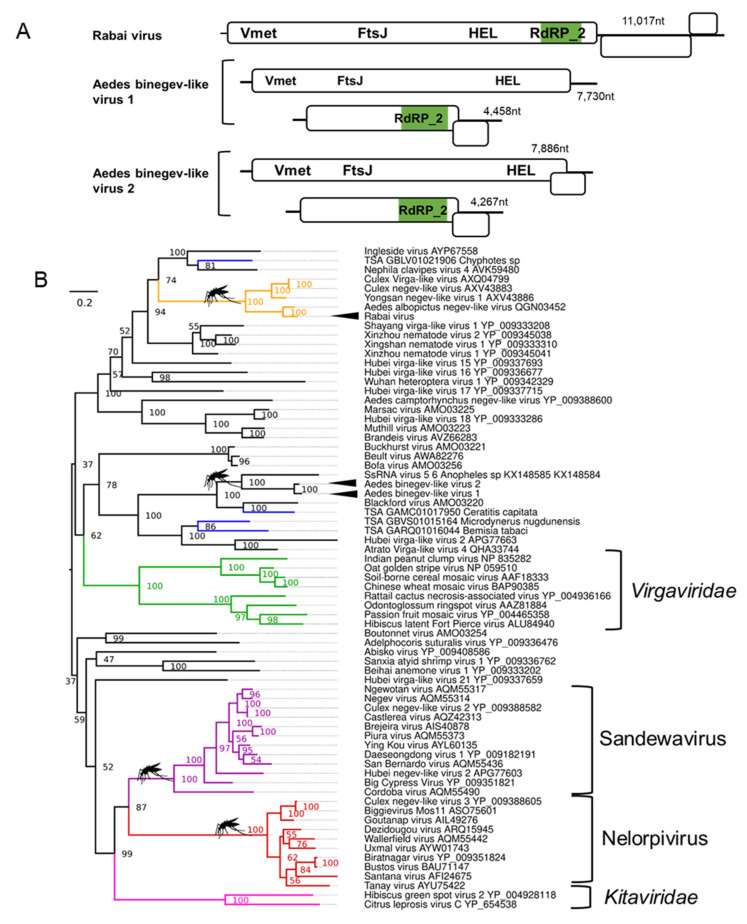
Novel positive-sense RNA viruses of *Ae. albopictus* and *Ae. aegypti.* (**A**) Genome organisation of Aedes negeviruses and bi-negeviruses. Vmet, Viral methyltransferase; FtsJ, Ftsj-like methyltransferase; HEL, helicase, and (**B**) phylogenetic placement of novel negeviruses based on concatenated MET, HEL, and RdRp_2 domains. Clades are coloured, indicating the mosquito nelorpivirus (red) and sandewavirus (purple) negevirus clades, as well as the *Virgaviridae* (green) and *Kitaviridae* (pink) groupings, a divergent mosquito negevirus clade is indicated in orange. Branch length indicates the number of amino acid substitutions per site. Consensus maximum-likelihood trees are produced using IQTree as per Figure 2.

**Figure 5 microorganisms-09-01653-f005:**
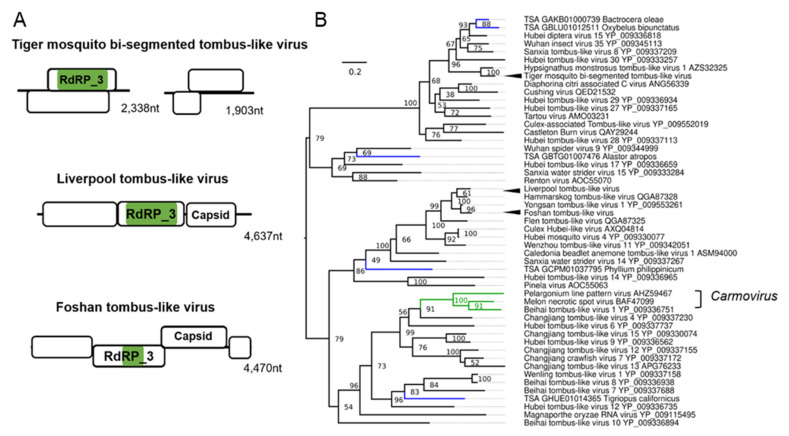
Novel positive-sense RNA viruses of *Ae. albopictus* and *Ae. aegypti.* Genome organisation of novel tombus-like viruses identified in this study (**A**) and phylogenetic placement (**B**) of viruses from aligned RdRp3 domains. Consensus maximum-likelihood trees were made using IQTree with the LG + G4 amino acid substitution model with 50,000 Ultrafast bootstraps with bootstrap support labelled on the node. Each TSA accession is shown in blue, members of the *Carmovirus* genus are indicated in green. Novel viruses are indicated with arrowheads. Branch length indicates the number of amino acid substitutions per site. For clarity, the tree is midpoint rooted.

**Figure 6 microorganisms-09-01653-f006:**
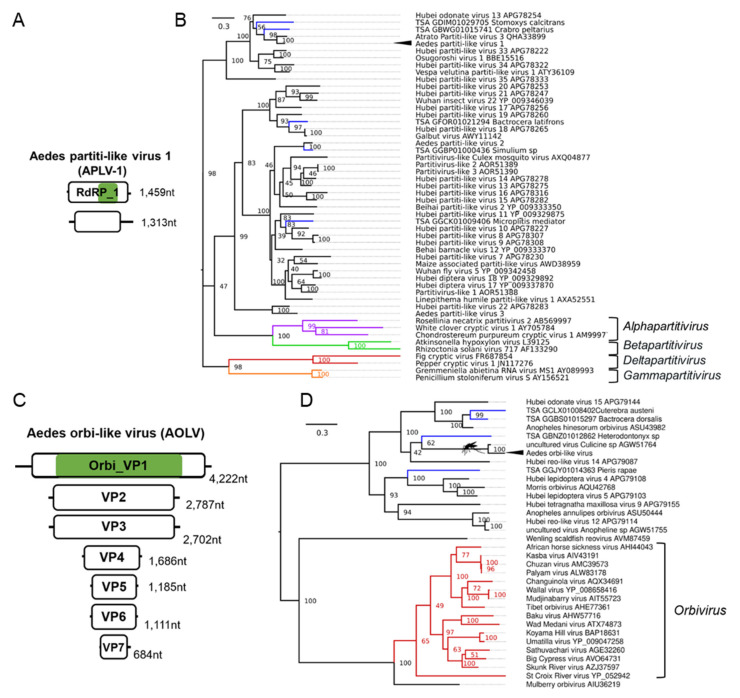
Novel dsRNA viruses of *Ae. aegypti*. Genome organisation (**A**), and phylogenetic placement (**B**) of Aedes partiti-like virus 1 (APLV-1) within the *Partitiviridae* family based on RdRp_1 domain alignment. ICTV accepted genus members from *Alphapartitivirus* (purple), *Betapartitivirus* (green), *Deltapartitivirus* (red), and *Gammapartitivirus* (orange). (**C**) Genome organisation of *Aedes* orbi-like virus (AOLV), and (**D**) phylogenetic position of AOLV based on aligned VP1 RdRp domain within the *Reoviridae* family. Members of the *Orbivirus* genus are indicated in red. Consensus maximum-likelihood trees were made with IQTree under the LG + G4 amino acid substitution model with 50,000 Ultrafast bootstraps. For clarity, the tree is midpoint rooted. Each TSA accession is shown in blue. Novel viruses are indicated with arrowheads. Nodes with an adjacent mosquito indicate virus clades identified from mosquito species. Branch length indicates the number of amino acid substitutions per site.

**Figure 7 microorganisms-09-01653-f007:**
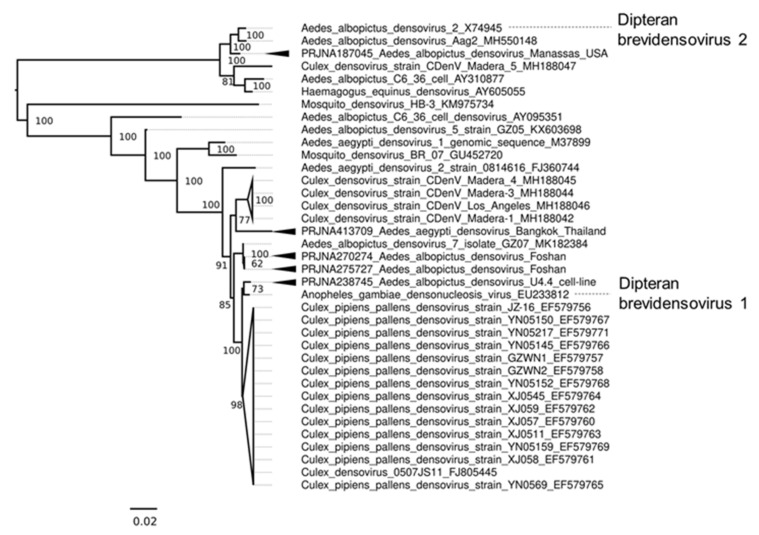
Phylogenetic relationship of Dipteran brevidensovirus strains identified in this study. Consensus tree constructed using NS1/NS2 and VP whole coding regions of virus genomes aligned using MAFFT. Ambiguous alignment regions were removed using GBLOCKS. Consensus maximum-likelihood trees were constructed with IQTree under the TIM2 + F+G4 nucleotide substitution model with 50,000 Ultrafast bootstraps. Bootstrapping support was conducted using 10,000 ultrafast bootstraps. Culex pipiens pallens densovirus and Culex densovirus nodes were collapsed for simplicity. The tree is midpoint rooted, and Genbank numbers corresponding to published virus strains are indicated on the label; novel strains of mosquito brevidensovirus are marked with arrowheads, along with the BioProject accession number.

**Figure 8 microorganisms-09-01653-f008:**
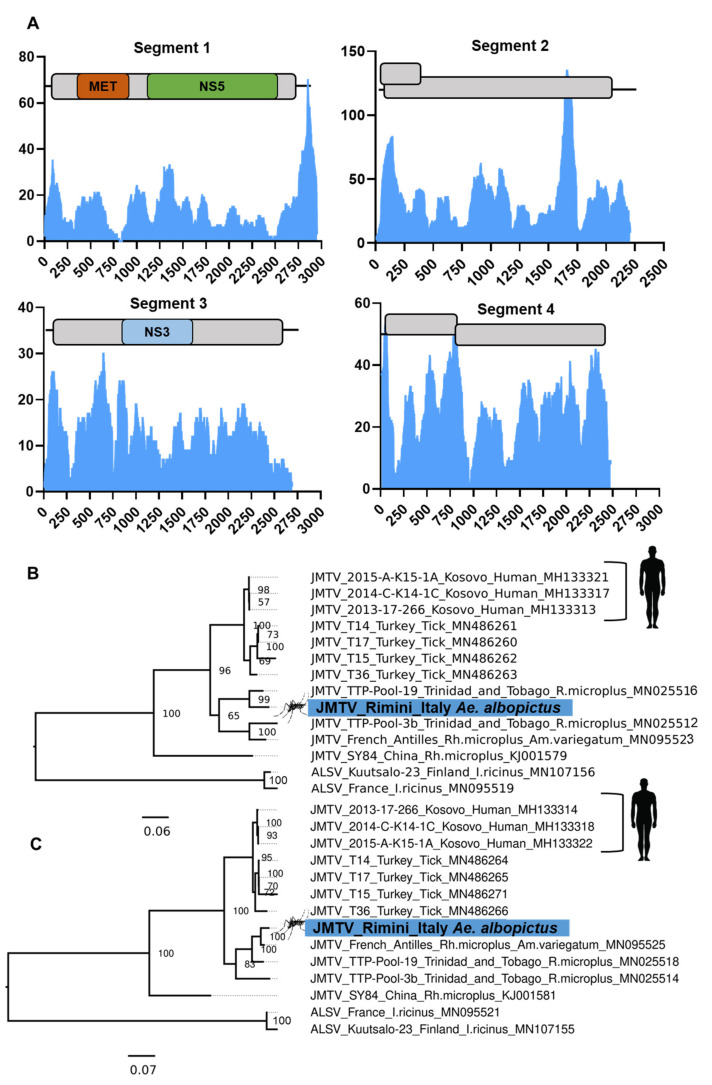
Evidence and phylogenetic placement of a Jingmen tick virus (JMTV) strain in the Rimini, Italy *Ae. albopictus* mosquito colony. (**A**) Coverage of the four segments of the Rimini JMTV strain originating from the head and fat body samples. Consensus trees constructed using (**B**) Segment 1 and (**C**) Segment 3 whole coding regions of virus genomes aligned using MAFFT and ambiguous alignment regions removed using GBLOCKS. Consensus maximum-likelihood trees were constructed with IQTree under the TIM2 + F+G4 nucleotide substitution model with 50,000 Ultrafast bootstraps. The trees are rooted on the Alongshan virus (ALSV) outgroup.

**Figure 9 microorganisms-09-01653-f009:**
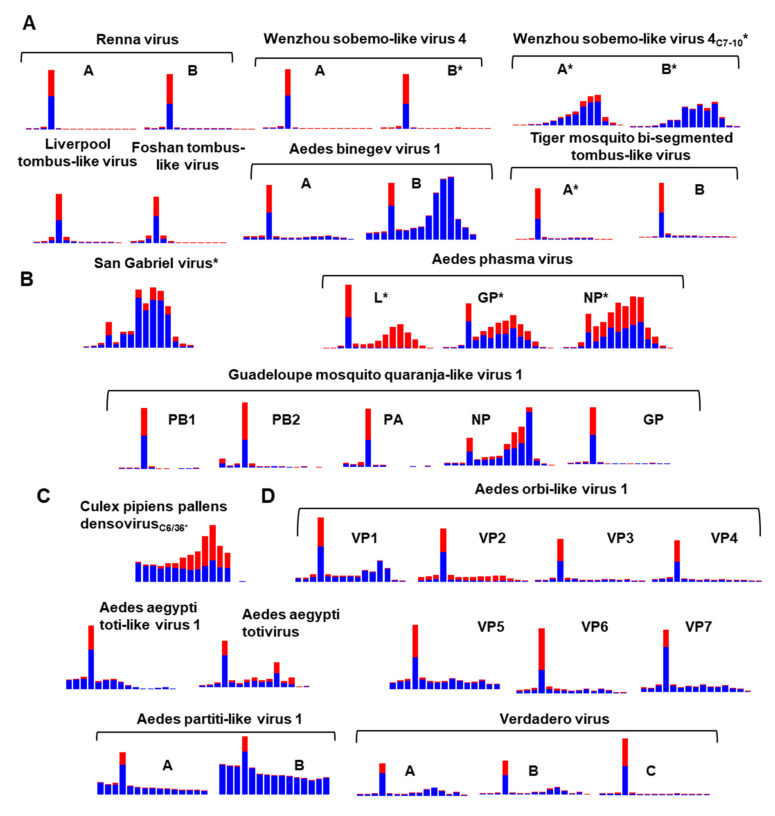
RNAi responses to *Aedes* viruses. Bar graphs represent extracted small-RNA reads (18–32 nt) mapped to the representative genomes of *Aedes* viruses presented in this study. Reads originating from the genomic sense are indicated in blue, and anti-genome sense reads are red. Virus segments that have both a characteristic piRNA signature (adenine at position 10, A_10_, for sense RNA; Uridine at position 1, U_1_ for antisense RNA), as well as overlapping 10 nt signature, are indicated with an asterisk (*). Viruses grouped by presumed Baltimore classification scheme with (**A**) IV: (+) ssRNA viruses, (**B**) V: (−) ssRNA viruses, (**C**) II: ssDNA virus, and (**D**) III: dsRNA viruses.

**Figure 10 microorganisms-09-01653-f010:**
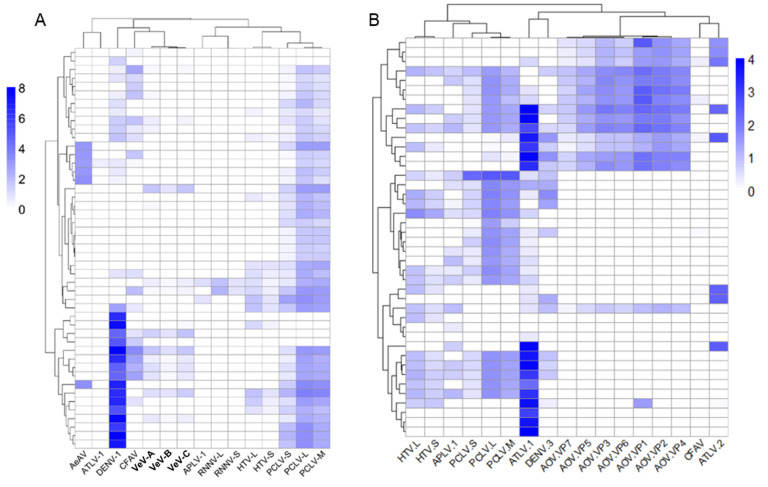
Abundance of viruses from individual *Aedes aegypti* mosquitoes. mRNA-enriched libraries prepared from midguts from Raquin et al. 2017 [119], (**A**), and whole mosquitoes from Koh et al. 2018 [108]. (**B**) Abundance is represented as log_10_(RPKM) mapped reads per million per kilobase. Hierarchical clustering was performed using Pearson complete distance measurement method. Heat map produced using pheatmap R package. RNNV: Renna virus, HTV: Humaita-Tubiacanga virus, APLV-1: Aedes partiti-like virus 1, VeV: Verdadero virus, DENV-1: dengue virus 1 (KDH0030A), DENV-3: dengue virus 3 (JN406515.1), AeAV: Aedes anphevirus, PCLV: Phasi Charoen-like virus, CFAV: Cell fusing agent virus, ATLV-1: Aedes toti-like virus 1, ATV: Aedes toti virus.

**Figure 11 microorganisms-09-01653-f011:**
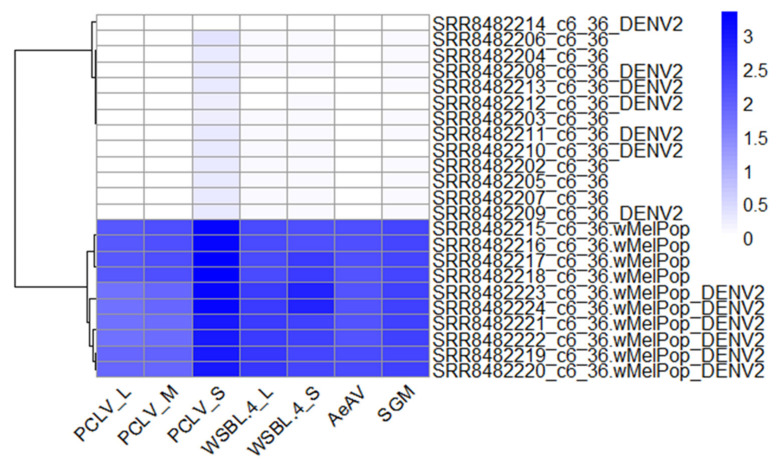
Evidence of ISV super-infection in *Wolbachia*-infected C6/36 cells. Heat map of data from Teramoto et al. 2019 [65]. Abundance is represented as log_10_(RPKM) mapped reads per million per kilobase. Hierarchical clustering was performed using Pearson complete distance measurement method. Heat map produced using pheatmap R package. Labels are: PCLV: Phasi Charoen-like virus, WSBL4: Wenzhou-Sobemo-like virus 4, AeAV: Aedes aegypti anphevirus, SGM: San Gabriel Mononegavirus.

**Table 1 microorganisms-09-01653-t001:** Novel and updated viruses of *Aedes* mosquitoes.

Provisional Virus Name; Host	Closest Relative; Genbank ID; Host	Classification: Order (O); Family (F); Genus (G)	Geographical Distribution
Formosus virus; *Aedes aegypti*	Isfahan virus; YP_007641386.1; *Phlebotomus papatasi*	O: *Mononegavirales*; F: *Rhabdoviridae*	**Laboratory colonies:** Bundibugyo, Uganda (U30) [54]
San Gabriel virus; *Aedes albopictus*	Wuhan ant virus; YP_009304559.1; *Camponotus japonicus*	O: *Mononegavirales*; F: *Rhabdoviridae*	**Laboratory colonies:** San Gabriel Valley, Los Angeles County USA [57,58]; Kawasaki, Kanagawa Prefecture, Japan [59]; **Wild-caught mosquitoes:** Yunnan, China [60]; Ticino, Muzzano, Switzerland [61] **Cell lines:** *Aedes albopictus* cells: RML-12.*w*MelPop [62], Aa23 cells and derivatives [63], C6/36.*w*MelPop [64,65] *Aedes aegypti* cells: Aag2.*w*MelPop [66,67]
Longgang virus (LMCV); *Aedes albopictus*	Shayang fly virus 1; YP_009300663.1; *Atherigona orientalis*	O: *Mononegavirales*; likely Chuvirus	**Laboratory colonies:** Longgang district, Shenzhen, China [68].
Aedes orthomyxo-like virus 2 (AOMV-2); *Aedes albopictus*	Whidbey virus; AQU42764.1; *Aedes dorsalis*	F: *Orthomyxoviridae*	**Laboratory colonies:** Foshan, China [69,70]; Italy, Rome [71]; Torres strait island, Australia [72]. **Wild-caught mosquitoes:** Zhejiang, China [73]
Rabai virus; *Aedes aegypti*	Yongsan negev-like virus 1; AXV43886.1; *Culex inatomii*	Unclassified Negevirus taxon related to F: *Virgaviridae*	**Laboratory colonies:** Rabai, Kenya (K2, K4) [54]
Aedes binegev-like virus 1 (AeBNV-1); *Aedes aegypti*	ssRNA virus-like 6 genomic sequence KX148585.1; *Anopheles gambiae*	Unclassified Negevirus taxon related to F: *Virgaviridae*	**Laboratory colonies:** Miami, Florida, USA [74]; Nova Iguaçu Rio de Janeiro, Brazil [75]; Bangkok, Thailand [76]; Key West & Orlando, USA [77]; Liverpool Colony [78,79]; Curepe, Trinidad [80] **Wild-caught:** Bangkok, Thailand [81]; Manatee County, USA [82]; Miami, USA [12].
Aedes binegev-like virus 2; (AeBNV-2) *Aedes albopictus*	ssRNA virus-like 6 genomic sequence KX148585.1; *Anopheles gambiae*	Unclassified Negevirus taxon related to F: *Virgaviridae*	**Laboratory colonies:** Longgang District, Shenzhen, China [68]; Manassas, USA [83].
Tiger mosquito bi-segmented tombus-like virus (TMTLV); *Aedes albopictus*	Culex mosquito virus 1; AXQ04816.1; *Culex sp.*	Related to the arthropod infecting F: *Nodaviridae* and F: *Tombusviridae*	**Laboratory colonies:** Gainesville (MRA-804), USA [84,85]; Phu Hoa, Binh Duong Province, Vietnam [59,86]; Manassas, USA [83,87,88,89,90]; Nice, France [91]; Wise County, Virginia, USA [92]; Foshan, China [93,94,95]; Kawasaki, Kanagawa Prefecture, Japan [59].
Liverpool tombus-like virus (LTLV) *Aedes albopictus,* *Aedes aegypti*	Hammarskog tombus-like virus; QGA87328.1 *Coquillettidia richiardii*	Related to the arthropod infecting F: *Nodaviridae* and F: *Tombusviridae*	**Laboratory colonies:** Liverpool strains [70,96,97,98,99,100,101], Poza Rica, Mexico [102]; Chetumal (CTM), Mexico [103]; Higgs White Eye (HWE) strain (Variant of Rex-D), Puerto Rico [104].
Aedes orbi-like virus (AOLV); *Aedes aegypti*	uncultured virus; AGW51764.1; *Culicine* sp.	F: *Reoviridae* G: *Orbivirus*	**Laboratory colonies:** Cairns, Innisfail, and North QLD Australia [105,106,107,108] **Wild-caught:** Cairns, Australia [81]
Aedes partiti-like virus 1 (APLV-1); *Aedes aegypti*	Hubei partiti-like virus 34; APG78322.1; Chinese land snail	F: *Partitiviridae*	**Laboratory colonies:** Rabai, Kenya (K2, K4, K14) [54]; Galveston, USA [109,110]; Tapachula, Mexico [111]; Cairns, Innisfail, Townsville, Australia [105,106,107,108,112]; Laos [113]; New Delhi, India [114]; Singapore [115,116]; Kamphaeng Phet, Thep Na Korn and Bangkok, Thailand [117,118,119]; Curepe, Trinidad [80,120] **Wild-caught:** Bangkok, Thailand [81], Cairns, Australia [81]; Manatee County, USA [82]; Miami, USA [12]; Les Abymes & Petit-Bourg, Guadeloupe [121]

## Data Availability

All virus assemblies that have not been directly sequenced by the authors from previous projects will be submitted to the Third-Party Annotation (TPA) database and made available for download after manual curation annotation from NCBI staff. All primary sequencing data are available from accessions listed in the Appendix A. The NCBI BioProject ID for this study is PRJNA720639.

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
