# Peer review of "Uncovering the Worldwide Diversity and Evolution of the Virome of the Mosquitoes Aedes aegypti and Aedes albopictus"

_microorganisms, 2021, doi:10.3390/microorganisms9081653_

Round 1

Reviewer 1 Report

In the last two decades, the discovery of insect-specific viruses has increased dramatically. The in the present study sequenced/identified ten novel viruses from call lines and mosquitoes from various sources. The study highlights the abundance of insect-specific viruses in mosquitoes and will facilitate future studies to investigate the role of these viruses in evolution of pathogenic arboviruses. 

Author Response

As far as we can see there are no specific comments to address. We appreciate the reviewer's time in assessing this manuscript. 

Reviewer 2 Report

Parry et al. performed a meta-analysis using published RNA-seq database to investigate the diversity of the virome from two representative mosquito species, Aedes aegypti and Aedes albopictus. They found ten new viruses from multiple virus families. And they revealed that there was significant variation of ISVs from same mosquito colony. The overall story is interesting. The manuscript is well written and the figures are organized. I only have several minor comments:

  1. Because ~3000 RNA-seq libraries was included in this analysis and you can image there should be huge variation on sequencing depth, data quality and sequencing platform et al. , I am wondering how to solve these issues when combine these dataset for further analysis.
  2. Some findings were based on the lab-adapted mosquito clones or cell lines. But cross contamination is not rare in the lab especially after long term adaption. Are there any methods to help determine whether this phenomenon also happens in the wild mosquitoes? For examples, some new viruses were identified in the lab-adapted clone. Are these viruses present in the wild mosquitoes?
  3. I’m not sure whether it is common to share the pipeline code on a public web like GitHub. If the pipeline code is deposited, more details can be found and other researcher can easily repeat the results.
  4. An abbreviation issue: “RdRP” should be “RdRp”.

Author Response

Parry et al. performed a meta-analysis using published RNA-seq database to investigate the diversity of the virome from two representative mosquito species, Aedes aegypti and Aedes albopictus. They found ten new viruses from multiple virus families. And they revealed that there was significant variation of ISVs from same mosquito colony. The overall story is interesting. The manuscript is well written and the figures are organized. I only have several minor comments:

1. Because ~3000 RNA-seq libraries was included in this analysis and you can image there should be huge variation on sequencing depth, data quality and sequencing platform et al. , I am wondering how to solve these issues when combine these dataset for further analysis.

We appreciate that it is difficult to normalise between libraries and make quantitative analyses of the combined data. On lines 764-771 we have detailed the limitations of our analyses:

"It is essential to appreciate that the RNA samples' processing, sequencing library preparation, instruments used, and read length of all data used between studies are vastly different. As such, it is challenging to normalise and draw quantitative comparisons between studies. We believe, however, that the analysis presented here reasonably incriminates samples as positively infected. However, we concede that given the limitations of the library preparations, viruses not containing poly-A tails or in huge abundance may be incorrectly characterised as “negative”. Given this limitation, we recommend screening mosquito samples using RT-PCR or RT-qPCR methods. "

We believe that we only have enough data to implicate colonies as infected not uninfected. Given the reasons that the reviewer states it's almost impossible to exclude infection in these colonies. We don't have specific ways to address the shortfall of this analysis however recommend screening using molecular techniques.

2. Some findings were based on the lab-adapted mosquito clones or cell lines. But cross contamination is not rare in the lab especially after long term adaption. Are there any methods to help determine whether this phenomenon also happens in the wild mosquitoes? For examples, some new viruses were identified in the lab-adapted clone. Are these viruses present in the wild mosquitoes?

We appreciate the line of inquiry and agree that cross-contamination is not rare in lab colonies and welcome further studies. As far as we can tell only one virus identified has not yet been isolated from wild mosquitoes. This study only offers genomic resources of novel viruses to explore the natural methods of transmission of these viruses in the wild. We believe extensive semi-outdoor settings and in vivo experiments are important to further our understandings of contamination or infection of these viruses.

3. I’m not sure whether it is common to share the pipeline code on a public web like GitHub. If the pipeline code is deposited, more details can be found and other researcher can easily repeat the results.

We appreciate that depositing the data and code on Github is an appropriate resource to ensure that the data is reproducible. A large portion of this project is modular and run on different graphical user interfaces and there is very little bash scripting to speak of. Therefore it is difficult to archive a version of this project at least retrospectively on Github. In the future, this will be done for all scripts that are open source but very difficult to do in this instance. We believe that the methods have been described in sufficient detail and also the data used clearly outlined in the supplementary files to ensure reproducibility. 

4. An abbreviation issue: “RdRP” should be “RdRp”.

We appreciate this has been pointed out by the reviewer and we have changed the abbreviation on line 304 where this was capitalized.